# TBP/TFIID-dependent activation of MyoD target genes in skeletal muscle cells

Barbora Malecova[1], Alessandra Dall'Agnese[1], Luca Madaro[2], Sole Gatto[1], Paula Coutinho Toto[1], Sonia Albini[1], Tammy Ryan[1], Làszlò Tora[3], Pier Lorenzo Puri[1,2]*

[1]Development, Aging and Regeneration Program, Sanford Burnham Prebys Medical Discovery Institute, La Jolla, United States; [2]Fondazione Santa Lucia - Istituto di Ricovero e Cura a Carattere Scientifico, Rome, Italy; [3]Cellular Signaling and Nuclear Dynamics Program, Institut de Génétique et de Biologie Moléculaire et Cellulaire, CU de Strasbourg, France

**Abstract** Change in the identity of the components of the transcription pre-initiation complex is proposed to control cell type-specific gene expression. Replacement of the canonical TFIID-TBP complex with TRF3/TBP2 was reported to be required for activation of muscle-gene expression. The lack of a developmental phenotype in TBP2 null mice prompted further analysis to determine whether TBP2 deficiency can compromise adult myogenesis. We show here that TBP2 null mice have an intact regeneration potential upon injury and that TBP2 is not expressed in established C2C12 muscle cell or in primary mouse MuSCs. While TFIID subunits and TBP are downregulated during myoblast differentiation, reduced amounts of these proteins form a complex that is detectable on promoters of muscle genes and is essential for their expression. This evidence demonstrates that TBP2 does not replace TBP during muscle differentiation, as previously proposed, with limiting amounts of TFIID-TBP being required to promote muscle-specific gene expression.

*For correspondence: lpuri@sbpdiscovery.org

**Competing interests:** The authors declare that no competing interests exist.

## Introduction

The process of RNA polymerase II (RNAPII) recruitment to protein coding genes is part of the assembly of the Pre-Initiation Complex (PIC) that is typically nucleated by TFIID-mediated recognition of core promoter sequences. The TFIID complex is composed of TATA-box binding protein (TBP) and TBP-associated factors (TAFs) (*Roeder, 1996*; *Tora, 2002*). Moreover, several TBP-related factors (TRFs) have been identified (*Akhtar and Veenstra, 2011*), expanding the repertoire and versatility of the basal transcription machinery (*Müller et al., 2010*; *Goodrich and Tjian, 2010*; *Müller et al., 2007*). While a number of events contribute to promote PIC assembly, including sequence-specific transcriptional activators, chromatin remodeling complexes and histone modifications, recent work has proposed that changes in the identity of PIC components is a key determinant of cell type-specific gene expression (*Hart et al., 2007*; *Deato and Tjian, 2007*).

In particular, it was recently suggested that during skeletal muscle differentiation, canonical TFIID is replaced by a complex consisting of TBP2 (also termed TRF3 = TBP-related factor 3; gene name Tbpl2 = TATA box binding protein like 2) and TAF3, which is required for activation of muscle genes (*Deato and Tjian, 2007*; *Deato et al., 2008*). Further work has revealed that the alternative core factor TAF3, which is known to recognize histone H3K4me3 - a histone modification associated with active promoters (*Vermeulen et al., 2007*; *van Ingen et al., 2008*; *Lauberth et al., 2013*; *Stijf-Bultsma et al., 2015*) - mediates interactions between MyoD, permissive chromatin and PICs at promoters of muscle-specific genes (*Deato et al., 2008*; *Yao et al., 2011*). This finding suggested that

**eLife digest** The muscles that allow animal's to move are built predominantly of cells called myofibers. Like other specialized cell types, these myofibers develop via a regulated set of events called differentiation. In adults, this phenomenon occurs when muscles regenerate after an injury, and new myofibers differentiate from so-called satellite cells that already reside within the muscles.

Differentiation is regulated at the genetic level, and the development of myofibers relies on the activation of muscle-specific genes. A gene's expression is typically controlled via a nearby regulatory region of DNA called a promoter that can be recognized by various molecular machines made from protein complexes. In most adult tissues, such regulatory machineries contain a complex called TFIID.

Previously it was reported that the TFIID complex was eliminated from cells during muscle differentiation, and that an alternative protein complex called TBP2/TAF3 recognizes and regulates the promoters of muscle-specific genes. However, Malecova et al. have now discovered that TFIID is actually present, albeit at reduced amounts, in differentiated muscles and that it drives the activation of muscle-specific genes during differentiation. Further experiments also showed that the TBP2 protein is not required for differentiation of muscle cells or for the regeneration of injured muscles, and is actually absent in muscle cells.

Further studies are now needed to explore how the TFIID-containing complex works with other regulatory protein complexes that are known to help make muscle-specific genes accessible to TFIID. It will also be important to study the relationship between the down-regulation of TFIID components and the activation of muscle-specific genes that typically occurs in mature myofbers. Together these efforts will allow the various aspects of gene regulation to be integrated, which will help provide a more complete understanding of the process of muscle differentiation.

the widely established paradigm of a universal and prevalent use of TFIID as a promoter-recognizing complex in a variety of somatic cells might undergo drastic change during development. Nevertheless this hypothesis has been challenged by the observation that TBP2 null (TBP2_KO = TBP2 knock out) mice do not display any skeletal muscle phenotype (*Gazdag et al., 2009*), indicating that TBP2 is dispensable for developmental myogenesis (see http://www.europhenome.org/databrowser/viewer.jsp?set=true&m=true&x=Both Split&ln=Tbp2&project=All&zygosity=All&m=true&l=10386). However, Gazdag et al. did not explore the integrity of the regeneration potential of skeletal muscles in TBP2 null mice during adult life, and thus have not ruled out the possibility that TBP2 can specifically promote the expression of muscle genes in muscle stem (satellite) cells (MuSCs) - the direct effectors of post-natal and adult skeletal myogenesis (*Chang and Rudnicki, 2014*).

MuSCs are usually found beneath the basal lamina of unperturbed muscles in a dormant state (quiescence) (*Mauro, 1961*), but are rapidly activated upon damage to give rise to new, differentiated myofibers through extensive epigenetic reprogramming (*Dilworth and Blais, 2011*; *Sartorelli and Juan, 2011*; *Moresi et al., 2015*). Two key molecular events that invariably coincide with MuSC activation and commitment toward myogenic differentiation are (i) the expression of skeletal muscle transcription master regulator MyoD (*Zammit et al., 2004*; *Tapscott, 1988*) and (ii) the activation of the p38 signaling (*Jones et al., 2005*). These pathways cooperate to promote SWI/SNF-mediated chromatin remodeling and to increase levels of the active chromatin mark H3K4me3 (*Bernstein et al., 2005*) at promoters of muscle genes (*Simone, 2004*; *Rampalli et al., 2007*; *Forcales et al., 2012*). Indeed, studies from the Rando group have revealed that in activated MuSCs the large majority of muscle gene promoters show enrichment in H3K4me3 (*Liu et al., 2013*). Collectively, the data described above suggest that the H3K4me3 chromatin mark can mediate TAF3 recruitment to MyoD-target genes in activated MuSCs. Since TAF3 has been proposed as a mediator of TBP2 recruitment to muscle gene promoters (*Deato et al., 2008*; *Yao et al., 2011*), these data provide the rationale for investigating whether TBP2 deficiency can impair post-natal and adult skeletal myogenesis.

## Results

### TBP2 ablation does not affect MuSC differentiation and muscle regeneration after muscle injury

To investigate the role of TBP2 in postnatal and adult skeletal myogenesis in vivo we assessed the muscle regeneration potential of TBP2 null mice (*Gazdag et al., 2009*) in response to injury. Notexin-mediated muscle injury triggers extensive muscle damage and stimulates adult muscle regeneration and post-natal myogenesis in vivo, during which damaged muscle is replaced by newly formed myofibers arising from activated MuSCs. Histological assessment of regenerating muscles showed a complete recovery of the muscle architecture, with abundant presence of centrally nucleated fibers in both wild type (WT) and TBP2 null muscles at 12 days post injury - a time point when muscle regeneration is near completion (*Figure 1A*). Both WT and TBP2 null mice contain similar numbers of Pax7 positive MuSCs in a sub-laminar position within unperturbed muscles (*Figure 1B* upper panel, 1C). This result indicates that there are no defects in MuSC ontogenesis in the absence of *Tbpl2* expression. Importantly, we detected comparable numbers of MuSCs in muscles of WT and TBP2 null mice 12 days after injury (*Figure 1B* lower panel, *1C*), indicating an intact capacity of adult MuSCs to proliferate, self-renew and differentiate during ongoing muscle regeneration in the absence of TBP2. Finally, we used Fluorescence Assisted Cell Sorting (FACS) to isolate MuSCs from skeletal muscles of WT and TBP2 null mice before and 12 days after notexin-mediated injury, and analyzed their intrinsic myogenic potential ex vivo Cultures of MuSCs from all conditions yielded a comparable number of Myosin Heavy Chain (MHC)-positive multinucleated myotubes (*Figure 1D,E*), demonstrating that MuSCs from TBP2 null muscles have the identical myogenic potential of MuSCs from WT mice as they can readily differentiate into myotubes with equal capacity upon exposure to differentiation conditions in vitro.

Our in vivo data on adult muscle regeneration, as well as the intact differentiation potential of *Tbpl2*-lacking MuSCs ex vivo demonstrate that TBP2 is dispensable for adult skeletal muscle differentiation.

### TBP2 is not expressed in myotubes

Our data demonstrating the lack of requirement of TBP2 for adult skeletal muscle regeneration prompted us to compare the RNA expression profile of *Tbp* and *Tbpl2* genes in MuSCs isolated from skeletal muscles of wild type mouse by FACS, and in the C2C12 myogenic cell line (*Blau et al., 1983*). *Tbp* expression was detected in both MuSC-derived myotubes and in C2C12 myotubes (*Figure 2A*). On the contrary, we could not detect *Tbpl2* expression in myotubes derived from MuSCs or C2C12s (*Figure 2A*). *Ckm* RNA expression in MuSCs and in C2C12 confirms that cells were differentiated into mytubes. As a control for *Tbpl2* RNA detection, we analyzed total RNA extracted from murine ovary tissue (*Figure 2A*), as previous work demonstrated the ovary-specific expression of TBP2 in mice (*Gazdag et al., 2009*). Independent analysis of publicly available RNA-seq data from C2C12 myoblasts and myotubes (*Trapnell et al., 2010*) and of our RNA-seq data from MyoD-converted human fibroblasts, further confirmed the absence of *Tbpl2* expression in skeletal myoblasts and myotubes (*Figure 2—figure supplement 1*).

As a further control of accuracy for detection of *Tbpl2* in muscle cells, we transfected C2C12 myoblasts with a murine *Tbpl2*-expressing plasmid and monitored the *Tbpl2* and *Tbp* expression in differentiated C2C12 myotubes by immunoblot analysis of total cell lysates of C2C12 myotubes and by RT-PCR analysis of RNA isolated from C2C12 myotubes (*Figure 2B,C*). We could detect the TBP2 protein (*Figure 2B*) and *Tbpl2* transcript (*Figure 2C*) in C2C12 myotubes only upon ectopic expression of *Tbpl2*. Importantly, the ectopic *Tbpl2* expression in C2C12s did not affect the formation of myotubes and the expression levels of muscle differentiation genes, such as *Myog* and *Ckm* (*Figure 2C*). The data we present here demonstrate that *Tbpl2* is not expressed during differentiation of skeletal myoblasts into myotubes.

### TBP is required for skeletal muscle differentiation

Since TBP levels were reported to be significantly decreased during differentiation of skeletal myoblasts into myotubes (*Deato and Tjian, 2007*; *Zhou et al., 2013*; *Li et al., 2015*), while TBP2 is absent in differentiating myotubes (data reported here), we tested whether lower amounts of TBP

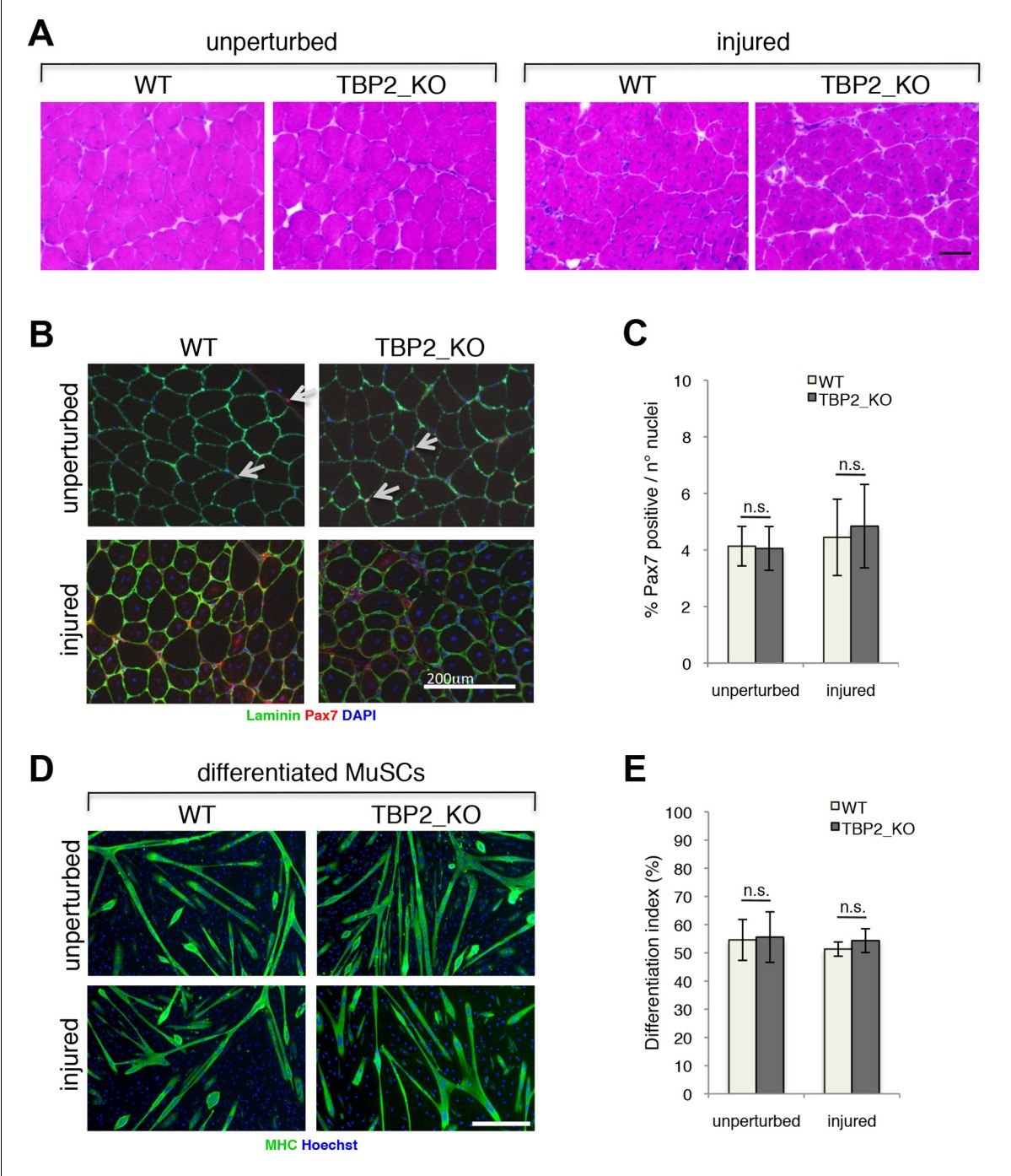

**Figure 1.** Regenerative potential and differentiation of MuSCs are intact in the absence of TBP2. (**A**) Cross-sections of unperturbed (left panel) and notexin injured (right panel) muscles were histologicaly assessed by Hematoxilin/Eosin (H&E) staining to evaluate the muscle architecture 12 days post notexin-mediated injury of the muscle. Size bar 100 μm. Representative image of muscles analyzed from two WT and two TBP2_KO mice are presented. (**B**) Representative images of cross-sections of unperturbed and of notexin injured muscles were immunolabeled with anti-laminin B2 (green) and anti-Pax7 (red) antibodies. Nuclei were counterstained with DAPI (blue). Pax7-positive MuSCs (Muscle Stem cells - Satellite cells) in sub-laminar position are indicated by an arrow. Size bar 200 μm. (**C**) Quantification of Pax7-positive MuSCs residing in sub-laminar position in B. Muscles from two mice have been analyzed and for each point, three different areas of the muscle were taken into account. Error bars represent a standard deviation among biological replicates. Student's *t* test was used for statistical analyses in C and E, n.s. - not significant. (**D**) MuSCs were isolated by FACS-assisted analysis as CD45-/CD31-/Ter119-/Sca1-/CD34+/alpha7intergrin+ from unperturbed and from notexin injured (12 days post injury) muscles. Isolated MuSCs were in vitro differentiated into myotubes, and immuno-labeled with anti-Myosin Heavy Chain (MHC) antibody (green) to monitor myotubes formation. The nuclei were counterstained with Hoechst 33,258 (blue). Images were taken by Olympus IX71 microscope with 10x objective.

*Figure 1 continued on next page*

*Figure 1 continued*
Representative images of differentiated MuSCs are presented. Size bar 50 μm. (**E**) Quantification of nuclei residing within MHC-positive differentiated MuSCs myotubes in D. Differentiation index is presented as% on nuclei within MHC-positive myotubes. Error bars represent a standard deviation between two independent MuSC isolations, each cultured and quantified in triplicate. Student's *t* test was used for statistical analyses in C and E, n.s. - not significant.

would be functional during muscle differentiation. We have effectively downregulated TBP protein levels in C2C12 myoblasts using an siRNA-mediated approach (*Figure 3A,C*), and exposed them to differentiation conditions. While C2C12s transfected with control siRNA readily differentiated into large multinucleated myotubes within 48 hr (*Figure 3A*, siCTR), C2C12 myoblasts with undetectable TBP protein levels failed to differentiate (*Figure 3A*, siTBP). We have quantified the differentiation index, the percentage of nuclei within myotubes of differentiated C2C12, to illustrate better the impaired differentiation potential of C2C12 in the absence of TBP (*Figure 3B*). mRNA analysis of skeletal muscle specific genes *Myog*, *Ckm* and *Acta1* demonstrates an impaired activation of the

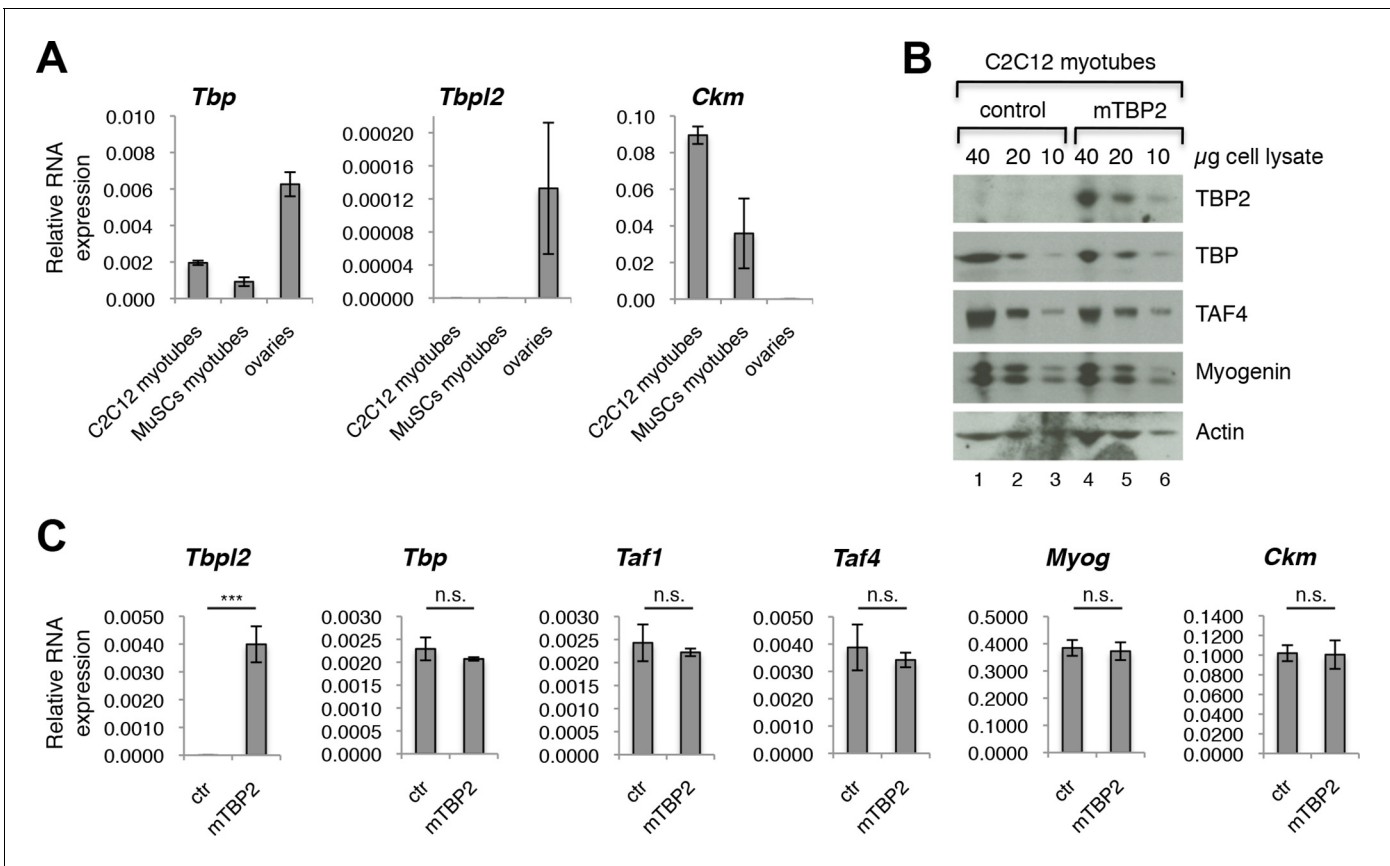

**Figure 2.** TBP2 is not expressed in myotubes. (**A**) RT-PCR analysis of RNA isolated from MuSC-derived myotubes, and from C2C12-derived myotubes. RNA isolated from murine ovaries was used as a control for *Tbpl2* expression (gene coding for TBP2 protein). Relative expression of indicated genes is presented as a fraction of *Gapdh* expression. Biological triplicates, error bars represent standard deviation. (**B**) Immunoblot analysis of the whole cell lysate of C2C12 myotubes exogenously expressing murine *Tbpl2* gene (mTBP2) or with a control plasmid. (**C**) RNA expression of indicated genes after exogenous expression of TBP2 in differentiated C2C12 myotubes. Biological triplicates, error bars represent standard deviation. Student's *t* test was used for statistical analyses (***p < 0.001, n.s. - not significant). Myotubes in all experiments shown in this figure were collected by careful trypsinization to avoid contamination with undifferentiated reserve cells (*Kitzmann et al., 1998*).

The following figure supplement is available for figure 2:

**Figure supplement 1.** RNA-seq data comparing Tbp and Tbpl2 gene expression in murine C2C12 and in human fibroblasts IMR90 converted by ectopic expression of MyoD.

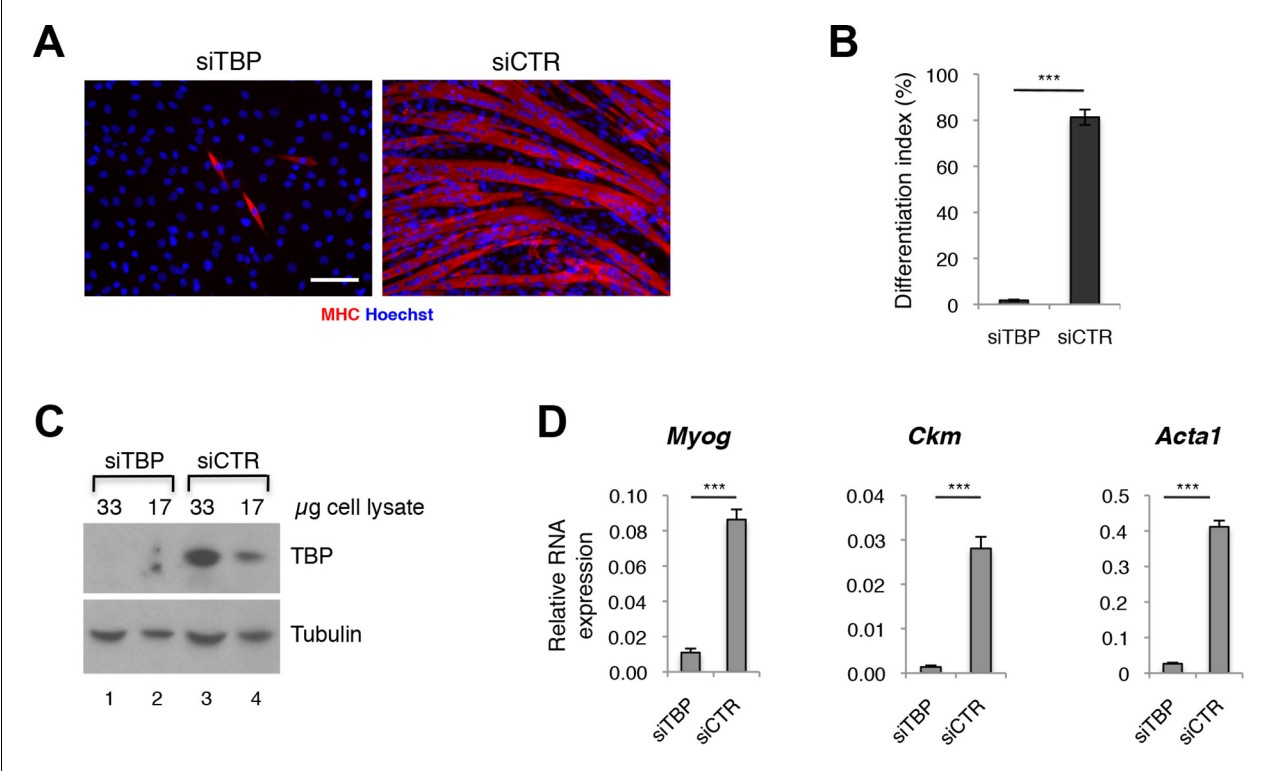

**Figure 3.** TBP is required for myoblast differentiation into myotubes. (A) Representative images of C2C12 myoblasts differentiation into myotubes upon TBP downregulation. TBP was downregulated using *Tbp*-targetting siRNA in C2C12 myoblasts. C2C12 myoblasts were differentiated into myotubes for two days, and immunolabeled with anti-Myosin Heavy Chain (MHC) antibody (red) to monitor myotubes formation. The nuclei were counterstained with Hoechst 33,258 (blue). Images were taken by Olympus IX71 microscope with 20x objective. Size bar 20 μm. (B) Quantification of nuclei residing within differentiated MHC-positive myotubes in A, represented as differentiation index. Error bars represent a standard deviation among three biological replicates. Student's *t* test was used for statistical analyses (***p<0.001). (C) Immunoblot analysis of the whole cell lysate of C2C12 to assess the TBP protein levels after siRNA-mediated silencing of TBP. (D) mRNA isolated from C2C12s grown in differentiation medium after siRNA-mediated downregulation of TBP was analyzed by RT-PCR for expression of skeletal muscle specific genes *Myog, Ckm* and *Acta1*. Relative expression of indicated genes is presented as a fraction of *Gapdh* expression. Error bars represent a standard deviation among three biological replicates. Student's *t* test was used for statistical analyses (***p < 0.001).

skeletal muscle program in the absence of TBP (*Figure 3D*). Thus, although TBP levels are reduced in myotubes compared to myoblasts under physiological conditions (*Deato and Tjian, 2007*), near complete TBP elimination impairs muscle differentiation. We conclude that TBP is required for skeletal muscle differentiation.

## The TFIID complex is expressed in myoblasts and myotubes

The requirement of TBP for skeletal muscle differentiation and the absence of TBP2 in skeletal myotubes suggest that the TFIID complex may be present in myogenic cells and may drive muscle gene transcription during myoblast differentiation into myotubes. To test this hypothesis we have set out to detect expression of TAF subunits of the TFIID complex in both cultured C2C12 and MuSC myoblasts and myotubes at the RNA and protein levels. We observe a downregulation of mRNA expression of the TFIID subunits TBP and TAFs in myotubes derived from both C2C12s (*Figure 4—figure supplement 1A*) and from MuSCs (*Figure 4—figure supplement 1B*), compared to undifferentiated cells (myoblasts). Next, we further analyzed the protein levels of TFIID subunits in myoblasts and in myotubes by immunoblot analysis (*Figure 4A*). Indeed, TFIID subunits were detected in C2C12 nuclear extracts prepared from both proliferating myoblasts and differentiated myotubes, with a concomitant downregulation of most TFIID subunits also at the protein level (*Figure 4A*), as previously observed (*Deato and Tjian, 2007*; *Zhou et al., 2013*). To demonstrate the purity of the myoblasts and myotubes isolated in our experimental conditions, we monitored the protein expression

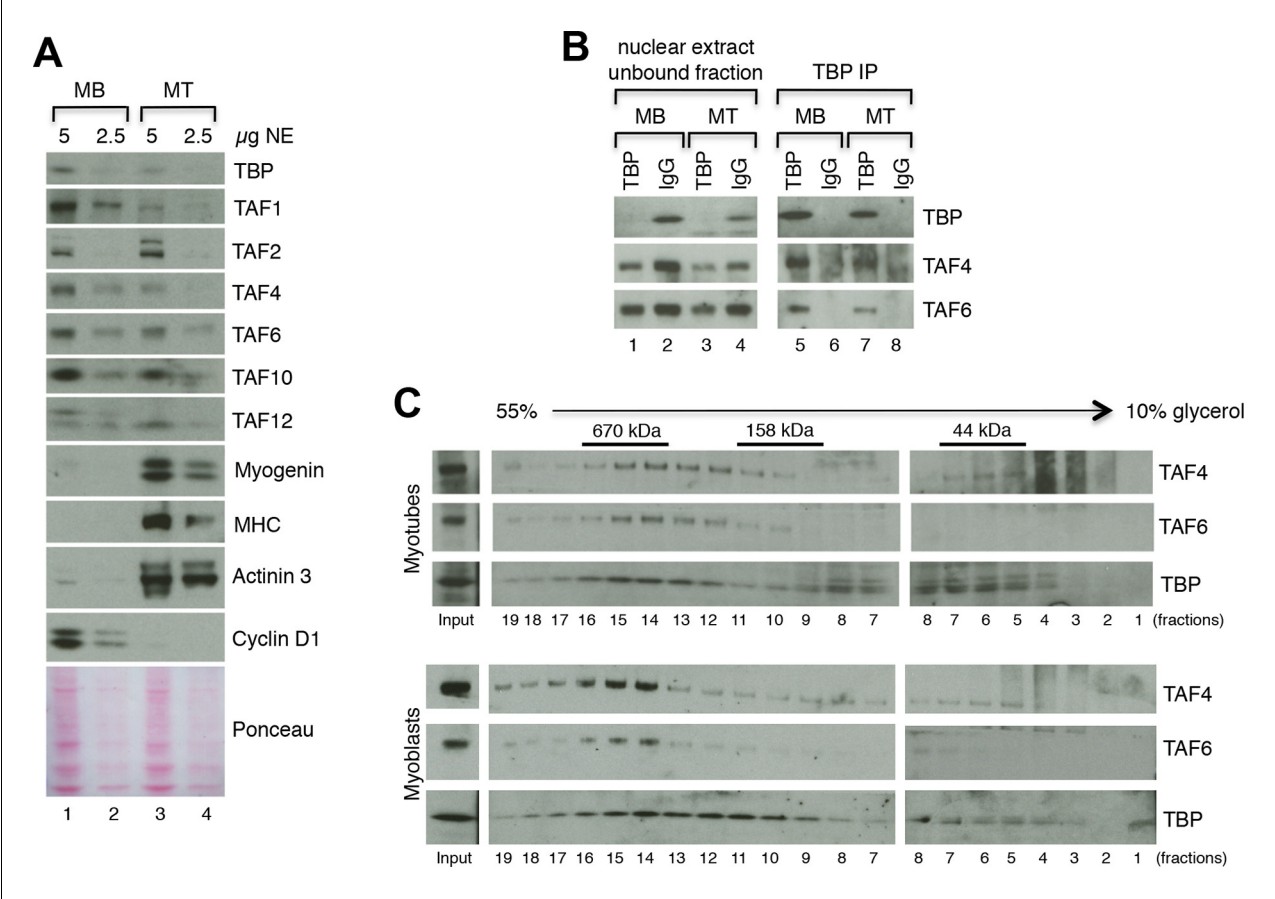

**Figure 4.** TFIID complex is present in proliferating myoblasts and differentiated myotubes. (A) Immunoblot analysis from nuclear extracts was used to determine protein levels of TFIID subunits TBP, TAF1, TAF2, TAF4, TAF6, TAF10, TAF12 expressed in proliferating myoblasts (MB) and differentiated myotubes (MT). Myoblast-specific expression of Cyclin D1 and myotube-specific expression of muscle markers Myogenin, MHC and Actinin 3 are used as markers of C2C12 differentiation. Ponceau staining of proteins on the immunoblot membrane was used as a loading control. Representative experiment is show. (B) Immunoprecipitation of endogenous TBP from C2C12 nuclear extract results in co-immunoprecipitation of TAF4 and TAF6 subunits of TFIID in both myoblasts (MB) and myotubes (MT). (C) 10–55% glycerol gradient size fractionation assay monitors co-fractionation of TFIID subunits TBP, TAF4 and TAF6 in high molecular weight fractions with a signal peaking in fractions corresponding to 0.6–1 MDa in both myoblasts (lower panel) and myotubes (upper panel).

The following figure supplement is available for figure 4:

**Figure supplement 1.** Evaluation of TFIID subunit expression on RNA level during C2C12 and Satellite cells (MuSCs) differentiation.

of the cell cycle protein Cyclin D1, which is typically expressed in myoblasts and downregulated during muscle differentiation (*Figure 4A*). We also analyzed markers of myogenic differentiation, such as Myogenin, Myosin Heavy Chain (MHC) and Actinin 3, (*Figure 4A*). Expression of these proteins is specific for either myoblasts (Cyclin D1) or myotubes (Myogenin, MHC, Actinin 3), and demonstrates the high purity of the cell populations used in our study, confirming that detection of TFIID subunits in myotubes is not due to contamination with undifferentiated myoblasts.

Next we analyzed the physical association between TBP and TAFs, as well as the integrity of the TFIID complex in C2C12 myoblasts and myotubes. Immunoprecipitation of endogenous TBP from nuclear extract prepared from C2C12 myoblasts and myotubes demonstrated an association of mTBP with mTAF4 and mTAF6 core subunits of TFIID in both proliferating and differentiated C2C12s (*Figure 4B*), further demonstrating the presence of the TFIID complex in myoblasts as well as in myotubes. Size fractionation of nuclear extracts from C2C12 myoblasts or myotubes by glycerol gradient centrifugation revealed the presence of TFIID subunits TBP, TAF4 and TAF6 in a high molecular weight fraction of about 0.6–1 MDa, thus detecting the TFIID complex in both myoblasts

and myotubes (*Figure 4C*), at the same size previously described in human HeLa cells (*Demény, 2007*).

Altogether, our results are contrary to previously reported data (*Deato and Tjian, 2007*) and demonstrate that the TFIID complex is not replaced by TBP2 upon myogenic differentiation.

## TBP is recruited to MyoD target genes during differentiation

The requirement of TBP for skeletal muscle differentiation prompted us to further analyze the dynamics of the PIC assembly on muscle promoters during C2C12 myoblast-to-myotube differentiation. Upon exposure of myoblasts to differentiation cues, we detected recruitment of TBP and RNA polymerase II phosphorylated on Serine 5 of its carboxy-terminal domain CTD (RNAPII-S5P), the transcription initiating form of RNAPII (*Hengartner et al., 1998*; *Komarnitsky et al., 2000*), on muscle gene promoters regulated by MyoD (*Figure 5*). TBP and RNAPII-S5P were not detected on these promoters in proliferating myoblasts, when the muscle genes are inactive (*Figure 5*), further supporting the evidence that TBP drives MyoD-mediated activation of muscle-specific gene transcription during muscle differentiation. TBP and RNAPII-S5P were present on the constitutively active *Gapdh* promoter in myoblasts and myotubes, but not on transcriptionally silent loci such as the *Sox2* promoter and the *Igh* enhancer (*Figure 5*). Thus, our data demonstrate the direct involvement of TBP in nucleating PIC assembly at MyoD-regulated skeletal muscle promoters during muscle differentiation.

## Discussion

In this work, we demonstrate that TBP, but not TBP2, is the essential component of the PIC that promotes muscle gene expression in differentiated skeletal muscles. We found that TBP2 is not expressed in myotubes derived from either C2C12 myoblasts, nor from primary mouse MuSCs. Consistently, TBP2-deficient mice displayed an intact regeneration potential, and MuSCs isolated from TBP2-deficient muscles differentiated into myotubes with the same efficiency as wild type MuSCs. This evidence firmly supports the conclusion that TBP2 does not replace TBP during muscle differentiation, as previously proposed by Deato et al. (*Deato and Tjian, 2007*).

We have observed downregulation of TFIID subunits at both the RNA and protein levels in differentiated myotubes, compared to proliferating myoblasts, as has previously been reported (*Deato and Tjian, 2007*; *Zhou et al., 2013*). Recently, a mechanism for ubiquitination-mediated downregulation of TBP protein in skeletal myotubes was reported (*Li et al., 2015*). Therefore, the

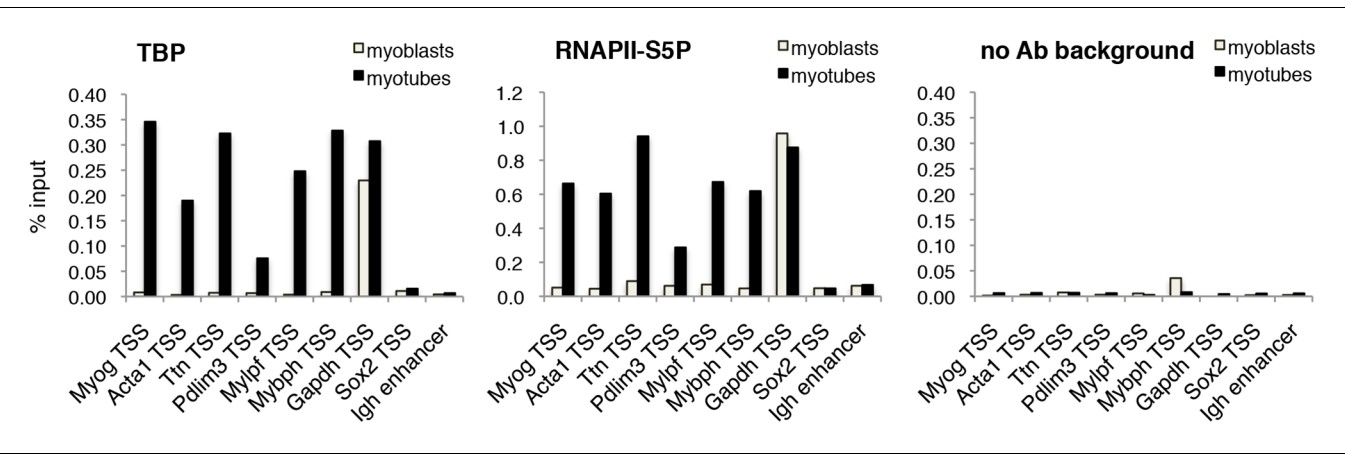

**Figure 5.** TBP is recruited to muscle genes upon muscle differentiation. Chromatin Immunoprecipitation (ChIP) assay in proliferating C2C12 myoblasts and differentiated C2C12 myotubes monitors recruitment of TBP and RNAPII-S5P components of PIC at promoters of skeletal muscle genes *Myog*, *Acta1, Ttn, Pdlim3, Mylpf* and *Mybph*. Transcriptionally active *Gapdh* promoter and inactive *Sox2* promoter and *Igh* enhancer were used as controls. Signal of relative recruitment is represented as a percentage of input DNA. Representative experiment of two independent biological replicate experiments is shown.

plausible scenario in skeletal muscle specific transcription is that limited TFIID helps to restrict the assembly of the transcription competent PIC at a selected subset of muscle specific genes.

Our findings that TBP/TFIID is functional in muscle progenitors and their differentiated progeny are in agreement with earlier studies showing that TBP was able to bind the MyoD promoter in a TATA-dependent manner (*Leibham et al., 1994*). It has been also shown that MyoD efficiently stimulated TFIID-dependent basal transcription (*Heller and Bengal, 1998*; *Dilworth, 2004*). Moreover, an endogenous interaction between MyoD and TBP in both myoblasts and myotubes was demonstrated, and in myotubes, MyoD and components of PIC TBP and TFIIB were found to be recruited to the proximal promoter of the *alpha-sarcoglycan* muscle gene promoter (*Delgado-Olguín et al., 2006*). The relevance of TBP-mediated activation of MyoD target genes has been recently emphasized by the finding that large polyglutamine (PolyQ) repeats in TBP causes muscle degeneration, by reducing the expression of muscle-specific genes, due to a decreased association of TBP with MyoD (*Huang et al., 2015*). This finding reiterates the importance of clarifying the identity of the proteins that compose the core promoter basal machinery required for muscle gene transcription in differentiating muscles, further extending the related implications to the pathogenesis of human muscular disorders.

## Materials and methods

### Animals and muscle injury

All protocols were approved by the Sanford-Burnham Medical Research Institute Animal Care and Use Committee. C57/BL6 TBP2_KO (TBP2 null; TBP2$^{-/-}$) mice were described previously (*Gazdag et al., 2009*). The mouse colony was maintained by breeding heterozygous females TBP2$^{+/-}$ with heterozygous males TBP2$^{+/-}$. Acute injury and muscle regeneration were induced in TBP2 null mice or in the wild type (WT) littermates by intramuscular injection of 10 µl of 10 µg/ml notexin (NTX, Latoxan, Valence, France) into the tibialis anterior (TA), gastrocnemius and quadriceps muscles. Mice were sacrificed 12 days post injury. Muscles were snap frozen in liquid nitrogen-cooled isopentane and cut transversally using a Leica CM 3050 S cryostat. Muscle regeneration was evaluated histologically on 10 µm cryo-sections of muscles.

### Histology

For Hematoxylin and Eosin staining (H&E, HT110-3, Sigma, St. Luis, MO), 10 µm muscle cryo-sections were stained in hematoxylin for 3 min and eosin Y for 2 min, according to the instructions. Stained cryosections were dried in ethanol and fixed in xylene, and slides were mounted with EUKITT mounting (O. Kindler GmbH & CO, Freiburg, Germany). Images were acquired using an inverted epifluorescent microscope (Nikon TE300, Tallahasee, FL), 10x objective lens. All images were processed using ImageJ. For Pax7 immunofluorescence experiments, 10 µm muscle cryosections were fixed in 1.5% paraformaldehyde for 15 min and permeabilized with 0.3% Triton-X. To avoid non-specific binding, muscle sections were blocked with 20% goat serum (Gibco by Thermo Fisher Scientific, Waltham) in PBS. Immunolabeling with a rat anti-Laminin B2 (05–206, Millipore, Billerica, MA, USA) antibody was performed overnight at 4°C, followed by labeling with secondary anti-rat antibody coupled with Alexa Fluor 488 (Molecular Probes, Thermo Fisher Scientific, Waltham, MA, USA). Next, the antigen retrieval was performed using the Antigen Unmasking Reagent (Vector Laboratories, Burlingame, CA, USA) at 100°C for 15 min. Specimens were next labeled with mouse anti-Pax7 antibody (PAX7, Developmental Studies Hybridoma Bank, Iowa City, IA, USA) in 20% goat serum and 0.3% Triton-X in PBS overnight at 4°C, followed by labeling with a secondary anti-mouse antibody coupled with Alexa Fluor 546 (Molecular Probes). Nuclei were visualized by counterstaining with DAPI (Molecular Probes, Thermo Fisher Scientific, Waltham, MA, USA). Images were captured using a Zeiss LSM 710 Multiphoton microscope equipped with Zen Software (Zeiss, Pleasanton, CA, USA) and subsequently edited in ImageJ.

### FACS isolation of Satellite cells (MuSCs)

Satellite cells were isolated from hind limb muscles of WT and TBP null mice, either unperturbed or 12 days post acute injury by fluorescence-activated cell sorting (FACS), as previously described (*Sacco et al., 2008*). Hind limb muscles were mechanically minced and enzymatically digested in

HBSS with $Mg^{2+}$ and $Ca^{2+}$ (Gibco by Thermo Fisher Scientific, Waltham, MA, USA) containing 0.2% (w/v) BSA and 10 Units/ml Penicillin + 10 µg/ml Streptomycin (Gibco by Thermo Fisher Scientific, Waltham, MA, USA), 2 mg/ml Collagenase A (Roche, Mannheim, Germany), 2.4 U/ml Dispase I (Roche, Mannheim, Germany), 10 µg/ml DNase I (Roche, Mannheim, Germany), 0.4 mM $CaCl_2$ and 5 mM $MgCl_2$ and 0.2% (w/v) Bovine Serum Albumin (BSA) for 90 min at 37°C. The cell suspension was filtered through a 100 µm nylon cell strainer (BD Falcon, Tamaulipas, Mexico), followed by filtration through 40 µm nylon cell strainer (BD Falcon). Cells were labeled with primary antibodies: CD31-PacificBlue (RM5228, Life Technologies by Thermo Fisher Scientific, Waltham, MA, USA), CD45-eFluor450 (clone 30-F11, eBioscience, San Diego, CA, USA), Ter119-eFluor450 (clone TER-119, eBioscience), Sca-1-FITC (clone E13-161.7, BD Pharmingen, by BD Biosciences, San Josa, CA, USA), CD34-APC (clone RAM34, BD Pharmingen) and α7integrin-PE (clone R2F2, AbLab, Vancouver, Canada) for 30 min at 4°C, in a HBSS buffer with $Mg^{2+}$ and $Ca^{2+}$ containing 0.2% (w/v) BSA and 10 Units/ml Penicillin + 10 µg/ml Streptomycin. Cells were washed and resuspended in HBSS with $Mg^{2+}$ and $Ca^{2+}$ containing 0.2% (w/v) BSA and Penicillin /Streptomycin. Dead cells were stained with Propidium Iodide (PI) (Sigma, St. Luis, MO, USA). Live PI-negative satellite cells were isolated as Ter119-/CD45-/CD31-/CD34+/α7-integrin+/Sca-1- cells. Fluorescence Activated Cell Sorting (FACS) was performed at the Sanford Burnham Prebys Medical Discovery Institute Flow Cytometry Core using a FACSAria instrument, and the data were analyzed by FACSDiva 6.1.3 software.

## Culture conditions for Satellite cells (MuSCs)

FACS isolated satellite cells were plated in laminin-coated tissue culture plates, in culture media containing 50% F10 media (Gibco by Thermo Fisher Scientific, Waltham, MA, USA), 50% DMEM media with low glucose (Gibco by Thermo Fisher Scientific, Waltham, MA, USA), containing 15% Fetal Bovine Serum (HyClone, Logan, UT, USA), 2.5 ng/ml basic fibroblast growth factor (bFGF, Prepo-Tech Inc. Rocky Hill, NJ, USA), and 10 Units/ml Penicillin + 10 µg/ml Streptomycin. After five days of culture, satellite cells were shifted to differentiation media containing DMEM High glucose (HyClone, Logan, UT, USA), 2% Horse serum (Gibco by Thermo Fisher Scientific, Waltham, MA, USA) and ITS liquid media supplement (Sigma, St. Luis, MO, USA) for an additional 3 days. Culture media was refreshed every two days.

## Immunofluorescence in Satellite cells (MuSCs)

Cultured differentiated Satellite cells were fixed in 4% paraformaldehyde in PBS for 15 min and permeabilized with 0.25% Triton-X in PBS for 10 min at room temperature. To avoid non-specific binding, cells were blocked with 20% goat serum in PBS for 30 min. Immunolabeling with anti-Myosin Heavy Chain (MHC) primary antibody (clone MF20, Developmental Studies Hybridoma Bank) in 5% goat serum overnight at 4°C. Secondary anti-mouse antibody coupled to Alexa Fluor 488 (Molecular Probes) was used to detect MHC signal in myotubes. Nuclei were counterstained with Hoechst 33258 (Life Technologies by Thermo Fisher Scientific, Waltham, MA, USA). Images were taken with Olympus fluorescent microscope with 10x magnification.

## C2C12 culture, differentiation and collection

C2C12 myoblasts were obtained from ATCC (Manassas, VA, USA) and cultured in DMEM High glucose media (growth media = GM) containing 15% Fetal Bovine Serum at low confluency. For differentiation, C2C12 cells were grown to 90% confluence and shifted to differentiation medium (DM) DMEM high glucose containing 2% horse serum and ITS Liquid Media Supplement for two days. Proliferating myoblasts were collected by scraping, while differentiated myotubes were collected by careful trypsinization with 0.05% Trypsin-EDTA (Gibco by Thermo Fisher Scientific, Waltham, MA, USA) to avoid contamination of myotubes by reserve cells (*Kitzmann et al., 1998*). After collection, C2C12 cells were washed with cold PBS containing 1 mM PMSF (Sigma, St. Luis, MO, USA). Collected C2C12 cells were frozen and stored at -80°C.

## Exogenous expression of mTBP2

C2C12 myoblasts were transfected with a pSG5-mTBP2 expressing plasmid (*Gazdag et al., 2007*) or a control empty plasmid using Fugene HD transfection reagent (Promega, Madison, WI, USA)

according to manufacturer's recommendation. Two days post-transfection, when C2C12 myoblasts were 90% confluent, cells were shifted to a differentiation medium (DM) DMEM high glucose containing 2% horse serum and ITS Liquid Media Supplement for two days. Differentiated myotubes were collected for RNA and protein analysis by careful trypsinization (*Kitzmann et al., 1998*) as described above.

## siRNA-mediated downregulation of TBP

C2C12 cells were transfected by forward transfection with 80 nM siRNA against *TBP* gene (siTBP, On Target plus Smart Pool L-041188-01, Dharmacon by Thermo Fisher Scientific, Waltham, MA, USA) or non-targeting siRNA (siCTR, D-002050-01-20, Dharmacon) using Lipofectamine RNAiMAX (Life Technologies by Thermo Fisher Scientific, Waltham, MA, USA) according the manufacturer's instructions. Two days post transfection, 90% confluent cells were shifted to differentiation media (DM) containing 2% horse serum with ITS supplement. Cells were collected after two days of differentiation in DM for further analysis by immunoblot and for RNA analysis.

## RNA analysis

RNA from C2C12 cells was extracted using RNeasy kit (Qiagen, Hilden, Germany) and the RNA from satellite cells (MuSCs) was extracted using miRNeasy Micro kit (Qiagen) following the manufacturer's protocol. cDNA was synthetized using QuantiTect Reverse Transcription kit (Qiagen) and analyzed by real-time quantitative PCR using SYBR Green PCR Master Mix (Applied Biosystems by Thermo Fisher Scientific, Waltham, MA, USA). Relative expression was calculated using $2^{-\Delta ct}$ method (*Livak and Schmittgen, 2001*). Primers used to detect gene-specific cDNA are listed in *Table 1* .

## Immunofluorescence in C2C12

C2C12 cells were fixed in 4% paraformaldehyde in PBS (Santa Cruz, Dallas, TX, USA) for 15 min and permeabilized with 0.5% Triton-X in PBS for 15 min at room temperature. To avoid non-specific binding, cells were blocked with 5% BSA in PBS for 30 min. Immunolabeling with anti-Myosin Heavy Chain (MHC) primary antibody (clone MF20, Developmental Studies Hybridoma Bank) in 5% BSA was done overnight at 4°C. Secondary anti-mouse antibody coupled to Alexa Fluor 555 (Molecular Probes) was used to detect MHC signal in myotubes. Nuclei were counterstained with Hoechst 33258 (Life Technologies by Thermo Fisher Scientific, Waltham, MA, USA). Images were taken with Olympus fluorescent microscope with 20x magnification.

## RNA-seq analysis of human IMR90 ectopically expressing MyoD

Human lung fibroblasts IMR90 (ATCC, Manassas, VA, USA) transfected with doxycycline inducible mouse MyoD were cultured in growth media (GM, DMEM high glucose containing 10% FBS). For the GM time point (proliferating converted myoblasts) MyoD expression was induced with 200 ng/ml of doxycycline (Sigma, St. Luis, MO, USA) 24 hr prior to the collection of cells. For the DM time point (differentiated myotubes), MyoD-expressing IMR90 cells were differentiated into myotubes for 72–96 hr in differentiation media (DM, DMEM high glucose containing 2% horse serum and ITS supplement). RNA was extracted using trizol (Ambion by Thermo Fisher Scientific, Waltham, MA, USA) following manufacturer's instructions. PolyA mRNA library preparation was performed as previously described (*Jin et al., 2013*) and sequenced on the Hi-Seq2000 platform. Reads were mapped to reference human genome (hg19) using TOPHAT v1.4.0 (http://tophat.cbcb.umd.edu/).

## Preparation of total cell lysate

Cells were lysed in RIPA lysis buffer containing 50 mM Tris-HCl pH 8.0, 150 mM NaCl, 5 mM EDTA pH 8.0, 0.5% SDS, 1% NP40, 0.5% Sodium Deoxycholate, 1 mM PMSF and protease and phosphatase inhibitor cocktail (Roche, Mannheim, Germany). Cell lysate was syringed through 29G1/2 needle ten times on ice and diluted five fold with TBS (50 mM TRIS-Cl pH 7.5, 150 mM NaCl) containing protease and phosphatase inhibitor cocktail (Roche). Cell lysates were spun at 14,000 g for 10 min at 4°C and supernatant was saved as soluble protein extract. Protein concentration of soluble cell lysates was determined by BCA assay (Pierce by Thermo FIsher Scientific, Waltham, MA, USA).

**Table 1.** Squences of the primers used in this study.

**Mouse primers for RT-PCR**

| | |
|---|---|
| Gapdh-for | GCTCACTGGCATGGCCTTCCG |
| Gapdh-rev | GTAGGCCATGAGGTCCACCAC |
| Myog-for | GAGACATCCCCCTATTTCTACCA |
| Myog-rev | GCTCAGTCCGCTCATAGCC |
| Acta1-for | AOCGTTTCCGTTGCCCOOAG |
| Acta1-rev | GGAGAGAGAGCGCGAACGCA |
| Ttn-for | GACACCACAAGGTGCAAAGTC |
| Ttn-rev | CCCACTOTTCTTGACCGTATCT |
| Pdlim3-for | TGGGGGCATAGACTTCAATCA |
| Pdlim3-rev | CTCCGTACCAAAGCCATCAATAG |
| Mylpf-for | TTCAAGGAGGCGTTCACTQTA |
| Mylpf-rev | TAOCGTCOAGTTCCTCATTCT |
| Mybph-for | CAGCCACTAAGCCTGAACCTC |
| Mybph-rev | TCCAACACATAGCCTTGAAGC |
| Cyclin D1-for | ACTTCCTCTCCAAAATGCCAG |
| Cyclin D1-rev | GTOGGTTGOAAATGAACTTCAC |
| Ckm-for | AQTCCTACACQQTCTTCAAGO |
| Ckm-rev | AGGAAGTGGTCATCAATQAGC |
| Tbpl2-for | ATACCTGGACCTCTTCCTGOAT |
| Tbpl2-rev | CCACCAAGATGTGGATGAAAC |
| Tbp-for | CCAAGCGATTTGCTGCAGTCATCA |
| Tbp-rev | ACTTAGCTGGGAAGGCCAACTTCT |
| Taf1-for | ACAGGAACAGATGCAGACCTTCGT |
| Taf1-rev | AATCTCCTCCTCAGGCACACCAAA |
| Taf2-for | GGCCTTGGAAAAATTCCCCAC |
| Taf2-rev | GAAGCACGCTGACATCCTGA |
| Taf3-for | GACATTGATGCTGCGAAAGTGCGA |
| Taf3-rev | TCCCGCTTGCT7CTTTCTCGATCT |
| Taf4-for | AGTTCACACGGCAAAGAATCACGC |
| Taf4-rev | AACGCCGGCTCATCCTGTTACTTA |
| Taf5-for | TTTC6GACGAGTAAATTCGTTCT |
| Taf5-rev | CTCCTGCACGATGTTCCAGAT |
| Taf6-for | AAACTCAGCAATACTGTGTTGCC |
| Taf6-rev | TTCTGTCGTTTCCCCATGTGC |
| Taf8-for | CCGGGAAGTAAGCAATCCACT |
| Taf8-rev | GCTTTCTCGGCACTCTCAAATC |
| Tar9-for | TGCCGAAAGATGCACAGATGA |
| Taf9-rev | TGTTGTCACATATCOGAAGGC |
| Taf10-for | GAGGGOGCAATQTCTAACGG |
| Taf10-rev | TGTGTAATCCTCCAACTGCATC |
| Taf12-for | GGACAGQTQQTCGTCTCAG |
| Tat12-rev | TCATCAOCOATCTOTAGCAGC |

**Mouse primers for ChIP**

| | |
|---|---|
| Myoq TSS-for | GCTCAGGTTTCTGTGGCGTT |

*Table 1 continued on next page*

*Table 1 continued*

**Mouse primers for RT-PCR**

| | |
|---|---|
| Myog TSS-rev | CCAAC TGCTGGGTGCC AT |
| Acta1 T5S-for | GTGCCCGACACCCAAATA |
| Acta1 TSS-rev | AGGGTAGGAAGTGAGGCTT |
| Ttn TSS-for | CCTTCCTAACAGAGCCAATCAC |
| Ttn TSS-rev | TGTTTCCTATGCAATCCCTACAC |
| Pdlim3 TSS-for | CACTCGCAGCAGGQATAAAT |
| Pdlim3 TSS-rev | GAACCGGACAACCTACTTAGC |
| Mylpf TSS-for | CTCCAAGCAGATTCTCTTGCTTT |
| Mylpf TSS-rev | GGTAGOGC TAT CC T OAOCTAAT |
| Mybph TSS-for | GCCTGCCTTTATAAGCATQAAC |
| Mybph TSS-rev | GTOTCAAGCTGOAGTOTTTAAG |
| Gapdh TSS-for | AGQGCTGCAGTCCGTATTTA |
| Gapdh TSS-rev | AGOAGGGGAAATGAGAOAGG |
| Sox2 TSS-for | GATTGGCCGCCGQAAAC |
| Sox2 TSS-rev | CTCTTCTCTOCCTTOACAACTC |
| Igh enhancer-for | AACCACAGCTACAAGTTTACC |
| Igh enhancer-rev | AACCAGAAÇACCTGCAGCAGC |

## Nuclear extract (NE) preparation

Nuclear extracts from C2C12 myoblasts and C2C12 myotubes carefully trypsinized (*Kitzmann et al., 1998*) were prepared as described previously (*Dignam, 1983*). For NE preparation, C2C12 cells were thawed on ice and rinsed in 10 ml hypotonic buffer (50 mM Tris-Cl pH 8.0, 5 mM EDTA pH 8.0, 20% glycerol). Next nuclei were gently resuspended in Low Salt buffer (10 mM KCl, 20 mM Tris-HCl, pH 7.2, 0.2 mM EDTA, 1.5 mM MgCl$_2$, 10 mM β-mercaptoethanol, 20% glycerol). High Salt Buffer (1 M KCl, 20 mM Tris-HCl, pH 7.2, 0.2 mM EDTA, 1.5 mM MgCl$_2$, 10 mM β-mercaptoethanol, 20% glycerol) was added gradually in 10 steps while stirring until the final KCl concentration reached 0.4 M. Proteins were further extracted by incubation at 4°C for additional 30 min. Samples were spun at 20,000 g for 20 min at 4°C. Supernatant containing nuclear extract was snap frozen in liquid nitrogen and stored at -80°C. Protein concentration of nuclear extracts was determined by BCA assay (Pierce).

## Endogenous TBP immunoprecipitation

Nuclear extracts from C2C12 myoblasts and myotubes were diluted with BC buffer (20 mM Tris-HCl, pH 7.2, 0.2 mM EDTA, 1.5 mM MgCl$_2$, 10 mM β-mercaptoethanol, 20% glycerol, 0.5 mM PMSF and protease and phosphatase inhibitor cocktail [Roche]) to a final KCl concentration of 150 mM. Protein concentration was measured using a BCA assay (Pierce). Before immunopreipitation, 250 μl (200 μg proteins) of nuclear extracts were precleared with 50 μl protein G-coated magnetic beads (Life Technologies by Thermo Fisher Scientific, Waltham, MA, USA) for 1 hr at 4°C. After pre-clearing, nuclear extracts were incubated overnight at 4°C with 10 μg of anti-TBP antibody (3TF1-3G3, mouse monoclonal) (*Brou et al., 1993*) or with 10 μg of non-specific mouse IgG (sc-2025, Santa Cruz, Dallas, TX, USA) cross-linked to protein G-coated magnetic beads using dimethyl pimelimidate, in the presence of 125 Units Benzonase (Novagen by Merck Millipore, Billerica, MA, USA). Protein G-bound immunocomplexes were washed seven times with BC buffer containing 150 mM KCl, and once with BC buffer containing 300 mM KCl. Isolated immunocomplexes were eluted from beads by incubation of samples with 60 μl of 1 mg/ml TBP peptide (PA81, synthetized by GenScript, Piscataway Township, NJ, USA) in BC buffer containing 150 mM KCl for 3 hr on ice.

## Glycerol gradient size fractionation of protein complexes

3.6 ml of 10–55% glycerol gradient was prepared by sequential layering of ten aliquots of 360 µl of BC buffer (150 mM KCl, 20 mM Tris-HCl, pH 7.2, 0.2 mM EDTA, 1.5 mM MgCl2, 10 mM β-mercaptoethanol, 20% glycerol, 0.5 mM PMSF ) containing and 55, 50, 45, 40, 35, 30, 25, 20, 15 and 10% glycerol. Nuclear extract was dialyzed against BC buffer containing 150 mM KCl and 8% glycerol at 4°C for 3 hr and spun down at 14,000 g for 15 min. 500 µg of dialyzed C2C12 nuclear extract was loaded onto 3.6 ml of 10–55% glycerol gradient, and size fractionated by centrifugation at 45,000 rpm for 16 hr in SW55Ti rotor (Beckman Coulter, Carlsbad, CA, USA) at 4°C. 19 glycerol gradient fractions of volume 200 µl were collected and analyzed by immunoblot analysis. Gel filtration molecular weight standard (Biorad, Irvine, CA, USA) containing Thyroglobulin (670 kDa), Gamma-globulin (158 kDa), Ovalbumin (44 kDa), Myoglobin (17 kDa) and Vitamin B12 (1.35 kDa) was analyzed in parallel to assess the Mw of glycerol gradient fractions. Fractions containing Mw standard were analyzed by 4–12% Tris-Glycine-SDS PAGE (Life Technologies by Thermo Fisher Scientific, Waltham, MA, USA) followed by Coomassie staining of the gels (Bio-Safe Coomassie G-250 stain, Biorad).

## SDS-PAGE and immunoblot analysis

Proteins of either nuclear extract or total cell lysate were size separated on 4–12% gradient Tris-Glycine-SDS denaturing PAGE (Life Technologies) and transferred onto a 0.45 µm nitrocellulose blotting membrane (Protran Amersham, by GE Healthcare, Little Chalfont, United Kingdom). The membrane was blocked with 5% milk in TBST (50 mM Tris_HCl pH 7.5, 150 mM NaCl, 0.1% Tween 20) buffer for 1 hr and subsequently probed with indicated antibodies in 5% milk in TBST buffer. Immunoblot membranes were washed with TBST and probed with goat anti-mouse or goat anti-rabbit secondary antibodies conjugated with horseradish peroxidase HRP (Biorad, Irvine, CA, USA). The signal was revealed using ECL Western Blotting Substrate (Pierce by Thermo Flsher Scientific, Waltham, MA, USA). Antibodies used for immunoblot probing were as follows: anti-TBP (3TF1-3G3, mouse monoclonal) (*Brou et al., 1993*), anti-TAF1 (sc-17134, Santa Cruz), anti-TAF2 (3038, rabbit polyclonal) (*Trowitzsch et al., 2015*), anti-TAF4 (32TA-2B9, mouse monoclonal) (*Wieczorek et al., 1998*), anti-TAF6 (25TA-2G7, mouse monoclonal) (*Wieczorek et al., 1998*), anti-TAF10 (6TA-2B11, mouse monoclonal) (*Wieczorek et al., 1998*), anti-TAF12 (22TA-2A1, mouse monoclonal) (*Wieczorek et al., 1998*), anti-TBP2 (2TBP-2B12, mouse monoclonal) (*Gazdag et al., 2007*), anti-Actinin 3 (EP2531Y, Origene, Rockville, MD, USA), anti-Cyclin D1 (EP272Y, EMD Millipore), anti-Myogenin (clone F5D, Developmental Studies Hybridoma Bank), anti-Tubulin (clone DM1A, Sigma), anti-Actin (sc-8432, Santa Cruz), anti-Myosin Heavy Chain (MHC, clone MF20, Developmental Studies Hybridoma Bank).

## Chromatin immuno-precipitation (ChIP) assay

C2C12 myoblasts and myotubes were collected by careful trypsinization (*Kitzmann et al., 1998*) and crosslinked in suspension in 40 ml PBS containing 1% formaldehyde for 15 min at room temperature. Formaldehyde was quenched by 0.125 mM glycine for 5 min at room temperature. Crosslinked cells were washed with cold PBS containing 1 mM PMSF. To prepare chromatin, cells were lysed in a ChIP lysis buffer containing 50 mM Tris-HCl pH 8.0, 150 mM NaCl, 5 mM EDTA pH 8.0, 0.2% SDS, 1% NP40, 0.5% Sodium Deoxycholate, 1 mM PMSF and protease and phosphatase inhibitor cocktail (Roche). Chromatin was sheared to an average DNA fragment length of 500bp using a Misonix3000 sonicator, and diluted five times with the ChIP lysis buffer lacking SDS, to the final concentration of 0.04% SDS. Samples were centrifuged and the protein concentration of soluble chromatin was determined by BCA assay (Pierce). 200 µg of chromatin was used for immunoprecipitation with 10 µl anti-TBP antibody (3TF1-3G3 mouse monoclonal) (*Brou et al., 1993*) or 8 µl anti-RNAPII-S5P antibody (Active Motif, 39233, Carlsbad, CA, USA). Sample with no antibody was used as a background control. After overnight incubation of the chromatin with the antibodies at 4°C, the immunocomplexes were captured with 50 µl protein A (for RNAPII-S5P antibody) or with 50 µl protein G (for TBP antibody) magnetic beads (Life Technologies) for further 3 hr at 4°C. Magnetic bead-bound immunocomplexes were washed four times with 1 ml buffer containing 50 mM Tris-HCl pH 8.0, 150 mM NaCl, 5 mM EDTA pH 8.0, 0.1% SDS, 1% NP-40, 0.5% Sodium Deoxycholate, 1 mM PMSF, and protease and phosphatase inhibitors cocktail (Roche), followed by one wash with a 1 ml buffer containing 250 mM LiCl, 100 mM NaCl, 5 mM EDTA pH 8.0, 1% NP40, 1% Sodium Deoxycholate, and a

subsequent final two washes with 1 ml TE buffer. Immunocomplexes were eluted from the beads for 15 min at 65°C with buffer containing 50 mM Tris-Cl pH 8.0, 1 mM EDTA pH 8.0 and 1% SDS. Cross-linking was reversed by incubation of the ChIP isolated immunocomplexes and the input chromatin samples at 65°C overnight in presence of 0.2 M NaCl. After 0.2 mg/ml Proteinase K treatment of samples, DNA from immunoprecipitated samples as well as DNA from 1% input was purified by phenol/chloroform extraction and ethanol precipitation. 1/50 of the purified DNA was analyzed by real-time quantitative PCR using SYBR Green PCR Master Mix (Applied Biosystems). Primers used to detect locus-specific signal are listed in *Table 1*. The ChIP signal is expressed as the amount of immunoprecipitated DNA relative to the input DNA (% of input).

## Acknowledgement

We thank A Cortez from Flow Cytometry core for FACS assistance, L Wang from Animal Facility for mouse embryo implantation, Zhen Ye and Samantha Kuan from Bing Ren's lab for RNA-seq technical assistance, E Scheer for help in antibody purification and preparations and ME Torres-Padilla for helpful discussions. This work was supported by NIH grants R01AR056712, R01AR052779 and P30 AR061303 to PLP, by ERC Advanced Grant (Birtoaction, grant N° 340551), ANR (13-BSV6-0001-02, COREAC; and ANR-13-BSV8-0021-03, DISCOVERIID), EU FP7 (PITN-GA-2013-606806, NR-NET) grants to LT, by CIRM training fellowship TG2-01162 to BM, by Dutch Parent Project NL fellowship and AFM fellowship to LM, by American-Italian Cancer Foundation fellowship to SG, by NIH diversity supplement to 5 R01 AR052779 to PCT, and by American Heart Association fellowship to TR.

## Additional information

### Funding

| Funder | Grant reference number | Author |
|---|---|---|
| California Institute of Regenerative Medicine | TG2-01162 | Barbora Malecova |
| Parent Project Muscular Dystrophy | postdoctoral fellowship | Luca Madaro |
| AFM-Téléthon | postdoctoral felowship | Luca Madaro |
| American-Italian Cancer Foundation | postdoctoral fellowship | Sole Gatto |
| National Institutes of Health | diversity supplement to 5 R01 AR052779 | Paula Coutinho Toto |
| American Heart Association | postdoctoral fellowship | Tammy Ryan |
| European Research Council | 340551 | Làszlò Tora |
| European Commission | PITN-GA-2013-606806 | Làszlò Tora |
| Agence Nationale de la Recherche | ANR-13-BSV8-0021-03 | Làszlò Tora |
| National Institutes of Health | R01AR056712 | Pier Lorenzo Puri |
| National Institutes of Health | R01AR052779 | Pier Lorenzo Puri |
| National Institute of Arthritis and Musculoskeletal and Skin Diseases | P30 AR061303 | Pier Lorenzo Puri |

The funders had no role in study design, data collection and interpretation, or the decision to submit the work for publication.

### Author contributions

BM, Conceptualization, Investigation, Data Visualization, Writing - Original Draft, Review and Editing, Conception and design, Acquisition of data, Analysis and interpretation of data, Drafting or revising the article; AD, LM, TR, Investigation, Acquisition of data, Analysis and interpretation of data, Drafting or revising the article; SG, PCT, SA, Data Visualization, Acquisition of data, Analysis

and interpretation of data, Drafting or revising the article; LT, Conceptualization, Review and Editing, Resources, Funding Acquisition, Conception and design, Analysis and interpretation of data, Drafting or revising the article, Contributed unpublished essential data or reagents; PLP, Conceptualization, Writing - Original Draft, Review and Editing, Resources, Funding Acquisition, Conception and design, Analysis and interpretation of data, Drafting or revising the article, Contributed unpublished essential data or reagents

### Author ORCIDs
Làszlò Tora, http://orcid.org/0000-0001-7398-2250
Pier Lorenzo Puri, http://orcid.org/0000-0003-4964-0095

### Ethics

Animal experimentation: This study was performed in strict accordance with the recommendations in the Guide for the Care and Use of Laboratory Animals of the National Institutes of Health. All of the animals were handled according to approved institutional animal care and use committee (IACUC) protocols (#13-007) of the Sanford Burnham Prebys Medical Discovery Institute.

## Additional files

### Major datasets

The following datasets were generated:

| Author(s) | Year | Dataset title | Dataset URL | Database, license, and accessibility information |
|---|---|---|---|---|
| Dall'Agnese A, Gatto S, Ye Z, Ren B, Puri PL | 2015 | IMR90 overexpressing MyoD | http://datadryad.org/review?doi=doi:10.5061/dryad.7qk36 | doi:10.5061/dryad.7qk36 |
| Malecova B, Dall'Agnese A, Madaro L, Gatto S, Coutinho Toto P, Albini S, Ryan T, Tora L, Puri PL | 2016 | Data from: TBP/TFIID-dependent activation of MyoD target genes in skeletal muscle cells | http://dx.doi.org/10.5061/dryad.7qk36 | Available at Dryad Digital Repository under a CC0 Public Domain Dedication |

The following previously published dataset was used:

| Author(s) | Year | Dataset title | Dataset URL | Database, license, and accessibility information |
|---|---|---|---|---|
| Trapnell C, Williams BA, Pertea G, Mortazavi A, Kwan G, van Baren MJ, Salzberg SL, Wold BJ, Pachter L | 2010 | Transcript assembly and abundance estimation from RNA-Seq reveals thousands of new transcripts and switching among isoforms | http://www.ncbi.nlm.nih.gov/geo/query/acc.cgi?acc=GSE20846 | GSE20846 |

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
