## [Decision Letter]

Thank you for submitting your work entitled "TBP/TFIID-dependent activation of MyoD target genes in skeletal muscle cells" for consideration by *eLife*. Your article has been favourably assessed by Kevin Struhl (Senior Editor) and three peer reviewers, one of whom, Alan Hinnebusch, is a member of our Board of Reviewing Editors, and another is Marc Timmers.

The reviewers have discussed the reviews with one another and the Reviewing Editor has drafted this decision to help you prepare a revised submission.

As you will see from the edited reviews shown below, two of the reviewers had issues with the experiments in Figure 1 on muscle regeneration following injury and it is important that you present cross-sections early after injury to document that the starting amount of injury was similar for the WT and TBP2_KO muscles. We also ask that you analyze additional control mRNAs to achieve a set of at least 3 reference genes for the qRT-PCR analyses. We all felt that adding MS analysis of the TFIID complex in TBP immunoprecipitates would greatly strengthen the paper and increase its impact on the field; although we do not insist on it. Finally, it is important that you increase the rigor of statistical analyses of your results.

Reviewer #1:

Previous work from the Tjian group suggested that, during skeletal muscle differentiation, canonical TFIID is replaced by a complex consisting of TBP-like factor TBP2 and TAF3, which is required for activation of muscle genes. This hypothesis has been challenged by the observation that TBP2_KO mice do not display any skeletal muscle phenotype, indicating that TBP2 is dispensable for developmental myogenesis. However, it remained possible that TBP2 was needed for the regeneration potential of skeletal muscles and the expression of muscle genes in muscle stem (satellite) cells (MuSCs). The results here are at odds with the latter possibility in revealing no defect in adult muscle regeneration or the differentiation potential of MuSCs ex vivo for TBP2 knock-out animals. The authors go on to provide evidence that TBP2 is not even expressed during differentiation of skeletal myoblasts into myotubes, although it could be detected in murine overies, while TBP and TAFs that comprise convential TFIID are readily detected. TBP2 expression can be observed in myotubes transfected with TBP2, but it has no effect on expression of muscle genes. They further demonstrate that down-regulation of TBP by siRNA impairs differentiation into large multinucleated myotubes and impaired activation of skeletal muscle specific genes. Thus, TBP2 is dispensable while TBP is required for differentiation from myoblasts to myotubes. In agreement with previous findings, they do observe a downregulation TBP and at least certain TFIID TAFs on differentiation of myoblasts to myotubes, but the residual proteins appear to assemble into TFIID, based on coIP and co-sedimentation analyses, and they are found by ChIP at the promoters of muscle genes in myotubes, but not myoblasts, at levels comparable to those seen at constitutively expressed genes and in parallel with the increased occupancies of Pol II CTD-phosphorylated on Ser5 in myotubes. These findings provide strong evidence that TBP and TAFs that comprise conventional TFIID, rather than the TBP2/TAF3 complex, mediates the induction of muscle genes by MyoD on differentiation of myoblasts to myotubes. This conclusion is in accordance with the recent report that polyglutamine repeats in TBP evokes muscle degeneration by reducing the expression of muscle-specific genes, with decreased association of TBPwith MyoD.

Major comments:

"With abundant presence of centrally nucleated fibers in both wild- type (WT) and TBP2_KO muscles 12 days post injury – a time point when muscle regeneration is under completion (Figure 1)": The presence of fibers is not documented here, but needs to be.

Figure 1: What analysis was done to establish that injury actually occurred?

Figure 2: Error bars indicating standard deviation and number of replicates (n) are required.

Figure 3: It would be superior to quantify results from multiple replicate experiments. The statement "Indeed, TFIID subunits were detected in C2C12 nuclear extracts prepared from both proliferating myoblasts and differentiated myotubes, with a global downregulation of most TFIID subunits also at the protein level (Figure 4)" needs quantification of replicates to justify it, as the conclusion is not obvious for TAFs 2, 6, 10, and 12.

For all experiments in which results cannot be quantified from replicates, it is necessary to state in the legend how many times the experiment was conducted and gave similar results.

Reviewer #2:

The manuscript by Malecova and collaborators focuses on the transcriptional regulation of MyoD target genes. The authors refer to a publication by Deato and colleagues (Deato et al., 2008), which concluded that differentiation from myoblasts to myotubes involves the replacement of the canonical TFIID by a novel complex composed of TBP2 and TAF3. This finding has been questioned by the lack of skeletal or muscular deficits in the TBP2^-/-^ mice (Gazdag et al., 2009). Malecova and collaborators extend the characterization of skeletal myogenesis in TBP2^-/-^ mice demonstrating how TBP2 ablation does not affect muscle regeneration after injury. The authors document absence of TBP2 expression in both mouse myoblast cell line C2C12 and muscle stem cells (MuSCs). In addition, they show that reduced TBP expression impairs myoblast differentiation and expression of muscle-specific genes. Using both C2C12 and primary muscle stem cells they confirmed that TBP and the TFIID complex become expressed to lower levels during myogenic differentiation. ChIP assays reveal a direct involvement of TBP in the pre-initiation complex assembly at MyoD-regulated genes.

The data presented by Malecova and colleagues contradicts the previously proposed mechanisms by Deato et al. for muscle differentiation, which were obtained in the C2C12 cell model in a clear and convincing sequence of experiments. In addition, the present study extends to primary cells and to the TBP2 deletion mice. The manuscript is well written and the results of this study are discussed in a fair and balanced manner in light of the published literature on this subject. I believe that this work represents an important addition and clarification of the role of basal transcription factors in muscle differentiation.

Major comments:

1) In Figure 1, the authors show the regenerative and differentiation potential of MuSCs in the TBP2^-/-^ mice. To better show the results in Figure 1, the authors should include a cross-section of notexin-injured muscles at an early time point (e.g. day 3). With this comparison, the authors can demostrate the difference between complete fiber breakdown observed at day 3 and the regeneration occurred at day 12.

2) The full sets of qRT-PCR have been quantified using GAPDH only as reference gene. The usage of more than one single reference is not sufficient. Many metabolic genes are altered upon myogenic differentiation. Therefore, additional reference genes should be used to come to a set of at least 3 reference genes, for example by inclusding HPRT1, HMBS, SDHA, PPIA, YWHAZ and/or B2M.

3) The authors detect the presence of the TFIID complex in myoblasts and myotubes with immunoblotting for TBP and a limited set of TBP-associated factors. Subsequently, they evaluate the integrity of the TFIID complex using a glycerol size fractionation. It would be better to investigate the TFIID complex by sensitive mass spectrometry of TBP immunoprecipitates to determine the precise composition of TFIID in myotubes.

Reviewer #3:

In this manuscript the authors present experiments designed to test whether TBP or TBP2 is required for muscle cell gene expression. The motivation for the experiments rests on previous conflicting studies. First, Deato et al. (2007, 2008) suggested that TBP2 is required for muscle cell gene expression, replacing TBP/TFIID with an alternative complex. However, Gazdag (2009) constructed a TBP2 homozygous null mouse and showed that it was viable, that TBP2 expression was only detectable in ovaries, that skeletal muscle in the TBP2^-/-^ mice was indistinguishable from that of control animals, and that the expression of myogenic factors was unaffected in the TBP2^-/-^ null mouse.

In the current work, the authors examine the issue in greater detail, testing whether TBP2 might be required in muscle stem cells, an aspect of muscle cell function untested by Gazdag (2009). Their work presents convincing data that TBP2 is not required for expression of muscle genes in these cells, that TBP2 is not detectable in these cells, and that TBP/TFIID is present and necessary for muscle gene expression. Thus, their work agrees with and extends the previous analysis of Gazdag (2009). The experiments are solid and the conclusions are reasonable.

---

## [Author Response]

*As you will see from the edited reviews shown below, two of the reviewers had issues with the experiments in Figure 1 on muscle regeneration following injury and it is important that you present cross-sections early after injury to document that the starting amount of injury was similar for the WT and TBP2_KO muscles.*

The reviewers pointed to an issue that deserves an important premise. Healthy, unperturbed skeletal muscles are composed of myofibers whose nuclei are typically located at the periphery. Intramuscular injection of myotoxins, such as notexin, is widely used to induce regeneration (1, 2, 3, 4) and is typically ensued by an early phase of muscle degeneration characterized by myofibers necrosis, followed by inflammatory infiltration and activation of satellite cells. These events occur during the first 3-4 days post injury 3 and prevent an accurate measurement of the cross-sectional area (CSA) of myofibers, as the injured muscle area contains high amount of cellular infiltrate, with the large majority of myofibers being destroyed and/or deformed (see Figure 6, 3 days post injury). As such, CSA cannot be evaluated during the initial stage of regeneration. Starting from day 5 or 6 after injury, activated satellite cells begin to fuse into new myfibers and/or fuse with the pre-existing damaged myofibers (3). This regeneration process is typically completed around day 10 post myotoxin injection (Figure 6), resulting in the restoration of the original muscle architecture. Well-established hallmarks of regenerating muscle at this stage are: i) the presence of newly regenerating myofibers with a smaller CSA and ii) central nucleation3 (Figure 6, 10 days post injury). Thus, in our experiments we have considered these two hallmarks to show that WT and TBP2_KO muscles were undergoing comparable regeneration, reflecting similar extent of injury. Indeed, we reasoned that failure to regenerate or weaker regeneration process would invariably lead to absence or reduction in number of centrally nucleated fibers by day 12 post-injury. On the other hand, failure to complete the regeneration process can be detected by the persistence of the inflammatory infiltrate at day 12 post-injury.

Author response image 1.Injury decreases the CSA in regenerating myofibers to the same extend in WT and TBP2KO.(**A**) Muscle histology: Hematoxilin & Eosin staining of WT muscle crosssections at different time point of regeneration. (**B**) Immunohistochemistry: cross-sections of WT and TBP2_KO muscles were stained with anti-collagen1 and anti-collagen-3 antibodies (red) to visualize clearly the myofibers. The nuclei were counterstained with DAPI. (**C**) Mean of all cross-section areas (CSA) quantified by ImageJ for each muscle was compared between injured and unperturbed muscle, the ratio is plotted. **D**) The centrally nucleated myofibers in 1B were quantified.**DOI:**
http://dx.doi.org/10.7554/eLife.12534.011

With this being taken into consideration, the inspection of injured muscles in our experimental system, revealed a comparable number of centrally nucleated fibers (see the quantification in Figure 6), and similar trends in changes of the CSA from unperturbed to day 12 post-injury muscles in WT and TBP2_KO mice (Figure 6). In both cases, the myofibers in injured muscles were 28% smaller compared to unperturbed muscles (Figure 6). Moreover, in both WT and TBP2_KO muscles the tissue architecture was restored 12 days post injury (Figure 1; Figure 6). This data support our conclusion that upon an initial injury, the regeneration ability of TBP2-deficient muscles is comparable to that of WT muscles.

While we provide these additional figures to the reviewers (and readers that will have access to these comments), we do not believe that they would add any further insight to the manuscript, and therefore we did not include them in the revised version of our work. However, we are happy to include them if the reviewers/editors deem it necessary.

*We also ask that you analyze additional control mRNAs to achieve a set of at least 3 reference genes for the qRT-PCR analyses.*

We indeed appreciate this comment from the reviewers. In an attempt to make our RNA analysis more rigorous, we have set out to normalize the RT-PCR data to the geometric average of 3 reference genes (5). We have designed primers for genes suggested by reviewer #2: *Sdha, Hprt, Ywhaz, Ppia, Hmbs,* as well as two additional ribosomal genes *Rpl10* and *Rpl0*. Primers for genes Sdha and Hmbs failed our quality control, therefore we excluded them from the analysis. Importantly, as common practice, we have reverse transcribed equal amounts of RNA into cDNA for myoblasts and myotubes. When comparing the expression of six housekeeping genes (*Gapdh, Hprt, Ywhaz, Ppia, Rpl10* and *Rpl0*), we have unfortunately noticed that majority of them are down-regulated in myotubes compared to myoblasts. We have plotted the raw Ct values for each analyzed sample to illustrate the differential expression of the analyzed potential reference genes during muscle differentiation (Figure 7, high Ct value = low expression). The most stable gene that is in our hands not differentially expressed during muscle differentiation, considering an equal total RNA amount, is *Gapdh*. For that reason we could not take into account any of the additional tested housekeeping genes for the data normalization purpose, and thus we could not use the normalization methods including geometric average of several reference genes (5), as we initially intended. Nonetheless this comparison with a panel of analyzed housekeeping genes that are down-regulated during muscle cells differentiation reinforced our initial conclusion that *Gapdh* is the most reliable reference gene for normalization of RNA expression.

Author response image 2.Comparison of expression levels of six houskeeping genes during myoblasts to myotubes differentiation.RNA was isolated from biological triplicates of C2C12 myoblasts and myotubes as described in themanuscript (see Figure 4—figure supplement 1). Identical amount of RNA was converted to cDNA for each sample. RNA expression for six houskeeping genes was evaluated by RT-PCR: *Gapdh*,*Hprt*, *Ywhaz, Ppia, Rpl10* and *Rpl0*. Ct values for each analyzed genes were plotted, the bar represents mean of three independent samples.**DOI:**
http://dx.doi.org/10.7554/eLife.12534.012

We all felt that adding MS analysis of the TFIID complex in TBP immunoprecipitates would greatly strengthen the paper and increase its impact on the field; although we do not insist on it.

Indeed, we are currently addressing this issue, within a more complex context (that is the relationship between PIC formation, SWI/SNF chromatin remodeling complex activity and H3K4 trimethylation), to possibly detect components of the basal transcription machinery that link SWI/SNF activity and activation of MyoD-regulated muscle promoters. While we acknowledge that the relationship between composition of the TFIID complex and SWI/SNF complex in myoblasts and myotubes builds directly on the discovery presented in this manuscript, we also note that it is part of an independent project that will need a substantial amount of work and that is clearly behind the timeframe of publication of the current manuscript. We will be more than glad to follow-up on this story, once ready, with a submission to *eLife*.

Finally, it is important that you increase the rigor of statistical analyses of your results.

Figure 1: We have clarified the number of replicates in the figure legend.

Figure 2: We have performed the RT-PCR experiment in triplicates, analyzing three independent skeletal muscle myotubes and ovary isolations and included the standard deviation error bars in the figure.

Figure 3: We have repeated the siTBP experiment in triplicates. We have immunolabeled the cells with anti-MHC (Myosin Heavy Chain) MF20 antibody to better visualize and quantify the extent of C2C12 myoblasts differentiation into myotubes, and the impairment of this process when TBP is downregulated (in the absence of TBP).

Figure 4: We have quantified the immunoblots of two independent nuclear extract preparations (Figure 8). We agree that it is easier to appreciate the downregulation of most of the TFIID subunits (except TAF2) on the protein level with the immunoblot quantification analysis. However, we have also noted that the antibodies against TBP, TAF2, TAF6, TAF10, MHC and Cyclin D1 do not react accordingly to the two-fold difference in amount of nuclear extract proteins on the immunoblot, since the quantified signals showed more than two-fold decrease in samples that were half amount of proteins. Therefore we believe that it would be incorrect and misleading to make any further quantitative conclusions about the extent of downregulation of the TFIID subunits on the protein level based on the immunoblot quantification. Therefore we are providing the quantification of immunoblot of two independent C2C12 myoblasts and myotubes preparation for an illustrative purpose in Figure 8.

Author response image 3.Downregulation of TFIID subunits on protein level - two independent nuclear extract (NE) preparations.(**A**) Original figure in the manuscript (Figure 4). (****B****) Immunoblot analysis of independent nuclear extract preparation. In panel B signals for TBP and TAF12 were not possible to quantify due to a high background:signal ratio. Immunoblot signal was quantified with ImageJ.**DOI:**
http://dx.doi.org/10.7554/eLife.12534.013

Figure 5: We have performed the ChIP experiment in C2C12 myoblasts and myotubes twice. In both independent experiments we have seen a similar extent of recruitment of TBP and RNAPII-S5P specifically at the promoters of muscle genes only in differentiated myotubes. Moreover, we have also performed ChIP experiments in our lab in differentiated C2C12 cells under the conditions of impaired skeletal muscle differentiation, when we downregulated Brg1 subunit of SWI/SNF chomatin remodeling complex (6, 7, 8). We observed that interference with the differentiation process leads to the targeted loss of recruitment of TBP and RNAPII-S5P on muscle promoters in our ChIP assays (data not shown).

We have performed Student’s t test statistical analysis on the data in Figure 1, Figure 2, Figure 3 and in Figure 4—figure supplement 1. The resulting p-values are indicated in the respective figures.

Figure 2—figure supplement 1: Since the main purpose of the RNA-seq data presentation in Figure 2—figure supplement 1 is to demonstrate the absence of TBP2 expression in skeletal myoblasts and myotubes, we think it is sufficient to present the figure in the paper as RNA-seq enrichment on the genome browser. Moreover, this visualization shows the quality of the RNA-seq data. Additionally, we have included the plots of normalized reads for genes relevant to this work in IMR90 conversion to skeletal muscle from an RNA-seq performed in our lab in collaboration with Dr. Ren's lab (Figure 9). The RNA-seq analysis of myoblasts and myotubes supports the RT-PCR analysis outcome presented on Figure 4—figure supplement 1 in regard to TBP downregulation during myoblasts to myotubes differentiation. According to the suggestion of Reviewer #1 regarding the Figure 2—figure supplement 1, we have modified the figure to better represent the differences in gene structure between murine (C2C12) and human (IMR90) cells to avoid confusion.

Author response image 4.Graphical representation of normalized RNA-seq data in human IMR90 fibroblasts overexpressing MyoD grown in growth media GM (myoblasts) or differentiated in differentiation media DM (myotubes).**DOI:**
http://dx.doi.org/10.7554/eLife.12534.014

References:

1. D’ Albis, A., Couteaux, R., Janmot, C., Roulet, A. & Mira, J. C. Regeneration after cardiotoxin injury of innervated and denervated slow and fast muscles of mammals. Myosin isoform analysis. Eur. J. Biochem. 174, 103–10 (1988).

2. Harris, J. B. & Johnson, M. A. Further observations on the pathological responses of rat skeletal muscle to toxins isolated from the venom of the Australian tiger snake, Notechis scutatus scutatus. Clin. Exp. Pharmacol. Physiol. 5, 587–600 (1978).

3. Chargé, S. B. & Rudnicki, M. A. Cellular and molecular regulation of muscle regeneration. Physiol. Rev. 84, 209–38 (2004).

4. Charrin, S. et al. Normal muscle regeneration requires tight control of muscle cell fusion by tetraspanins CD9 and CD81. Nat Commun 4, 1674 (2013).

5. Vandesompele J, Preter KD, Pattyn F, Poppe B, Roy NV, Paepe AD, Speleman F. Accurate normalization of real-time quantitative RT-PCR data by geometric averaging of multiple internal control genes. Genome Biology 2002; 3.

6. de la Serna, I.L., Ohkawa, Y., Berkes, C.A., Bergstrom, D.A., Dacwag, C.S., Tapscott, S.J., and Imbalzano, A.N. (2005). MyoD targets chromatin remodeling complexes to the myogenin locus prior to forming a stable DNA-bound complex. Mol Cell Biol 25, 3997–4009.

7. Forcales, S.V., Albini, S., Giordani, L., Malecova, B., Cignolo, L., Chernov, A., Coutinho, P., Saccone, V., Consalvi, S., Williams, R., et al. (2012). Signal-dependent incorporation of MyoD-BAF60c into Brg1-based SWI/SNF chromatin-remodelling complex. Embo J 31, 301–316.

8. Albini S, Coutinho Toto P, Dall'Agnese A, Malecova B, Cenciarelli C, Felsani A, Caruso M, Bultman SJ, Puri PL. (2015). Brahma is required for cell cycle arrest and late muscle gene expression during skeletal myogenesis. EMBO Rep. 2015 Aug;16(8):1037-50.